# Sarcopenia assessed by 4-step EWGSOP2 in elderly hemodialysis patients: Feasibility and limitations

**M. Luz Sánchez-Tocino** [1], **Blanca Miranda-Serrano**[1], **Carolina Gracia-Iguacel**[2], **Ana María de-Alba-Peñaranda**[3], **Sebastian Mas-Fontao**[4], **Antonio López-González**[5], **Silvia Villoria-González**[1], **Mónica Pereira-García**[3], **Alberto Ortíz**[2], **Emilio González-Parra**[2]*

**1** Fundación Renal Íñigo Álvarez de Toledo, Salamanca, Spain, **2** Servicio de Nefrología e Hipertensión, Fundación Jiménez Díaz, IIS-FJD UAM, Madrid, Spain, **3** Fundación Renal Íñigo Álvarez de Toledo, Madrid, Spain, **4** CIBERDEM, IIS-Fundación Jiménez Díaz UAM, Madrid, Spain, **5** Complejo Hospitalario Universitario A Coruña, A Coruña, Spain

* egparra@fjd.es

## Abstract

### Background

In 2019, EWGSOP2 proposed 4 steps to diagnose and assess sarcopenia. We aimed to quantify the prevalence of sarcopenia according to the EWGSOP2 diagnostic algorithm and to assess its applicability in elderly patients on hemodialysis.

### Methods

Prospective study of 60 outpatients on chronic hemodialysis aged 75- to 95-years, sarcopenia was assessed according to the 4-step EWGSOP2: <u>Find</u>: Strength, Assistance walking, Rise from a chair, Climb stairs, and Falls (SARC-F); <u>Assess</u>: grip strength by dynamometry (GSD) and sit to stand to sit 5 (STS5); <u>Confirm</u>: appendicular skeletal muscle mass (ASM) by bioimpedance; <u>Severity</u>: gait speed (GS), Timed-Up and Go (TUG), and Short Physical Performance Battery (SPPB).

### Results

The sequential four steps resulted in a prevalence of confirmed or severe sarcopenia of 20%. Most (97%) patients fulfilled at least one criterion for probable sarcopenia. The sensitivity of SARC-F for confirmed sarcopenia was low (46%). Skipping the SARC-F step increased the prevalence of confirmed and severe sarcopenia to 40% and 37%, respectively. However, 78% of all patients had evidence of dynapenia consistent with severe sarcopenia. Muscle mass (ASM) was normal in 60% of patients, while only 25% had normal muscle strength values (GSD).

### Conclusions

According to the 4-step EWGSOP2, the prevalence of confirmed or severe sarcopenia was low in elderly hemodialysis patients. The diagnosis of confirmed sarcopenia underestimated

---

**Data Availability Statement:** All relevant data are within the paper and its Supporting Information files.

---

**Funding:** This research received no external funding. EGP, SM and AO research groups are funded by Ministerio de Economia, Industria y competitividad: FIS/Fondos FEDER (PI16/01298, PI17/00257, PI18/01386, PI19/00588, PI19/00815, PI20/00487, PI21/01430, ERA-PerMed-JTC2018 (KIDNEY ATTACK AC18/00064 and PERSTIGAN AC18/00071, ISCIII-RETIC REDinREN RD016/0009), and Sociedad Española de Nefrología, Comunidad de Madrid en Biomedicina B2017/BMD-3686 CIFRA2-CM. All authors want to thank Fundación Renal Íñigo Álvarez de Toledo (FRIAT) for it support to renal research in Spain. The funders had no role in study design, data collection and analysis, decision to publish, or preparation of the manuscript.

**Competing interests:** The author(s) received no specific funding for this work. The authors have declared that no competing interests exist.

the prevalence of dynapenia consistent with severe sarcopenia. Future studies should address whether a 2-step EWGSOP2 assessment (Assess-Severity) is simpler to apply and may provide better prognostic information than 4-step EWGSOP2 in elderly persons on hemodialysis.

## 1. Introduction

Sarcopenia, defined as the loss of muscle mass and strength, has a negative impact on quality of life, morbidity and mortality [1]. Aging is associated with muscle mass loss, which in turn increases dependence and frailty [2]. The concept of sarcopenia is encompassed within the broader concept of frailty. Frailty refers to weakness, poor resistance and energy, slowness and scarce physical activity [3]. It is present in 7% of the elderly and in 42% of dialysis patients, and is associated with a 2.6-fold higher risk of mortality and 1.4-fold higher risk of hospitalization, regardless of age, comorbidity, and disability, as well as with an increased risk of disability, falls and institutionalization [4].

The reported variability of the prevalence of sarcopenia depends in part on the diagnostic criteria applied. In 2010, The European Working Group on Sarcopenia in Older people (EWGSOP) agreed on consensus diagnosis and assessment criteria for sarcopenia [1] which were recently revised and updated to generate the 4-step EWGSOP2 [5]. EWGSOP2 includes four steps: 1) Find, based on clinical suspicion or a screening questionnaire: Strength, Assistance walking, Rise from a chair, Climb stairs, and Falls (SARC-F) [6]; 2) Assess, that establishes the diagnosis of probable sarcopenia by assessing strength in upper and lower limbs; 3) Confirmation of sarcopenia by quantifying muscle mass by bioimpedance analysis (BIA) or more sophisticated and expensive techniques; 4) Severity, that assesses the ability to perform certain physical tests. Thus, once sarcopenia has been confirmed by muscle mass assessment, severity is determined by assessing its functional impact, which is related to the degree of dynapenia, understood as the age-associated loss of muscle strength that is not caused by neurologic or muscular diseases [7].

In Spain, the mean age at dialysis initiation is 65 years and the prevalence of older (over 75 years) age is high among dialysis patients [8]. However, there is scarce information on the prevalence of sarcopenia in the very old on hemodialysis, and the feasibility of performing EWGSOP2 in routine clinical practice and the information obtained from EWGSOP2 had not been previously assessed in this population [9]. The aim of the present study was to quantify the prevalence of sarcopenia according to the EWGSOP2 diagnostic algorithm and to assess its applicability in elderly patients on hemodialysis. Additionally, we assessed the association of EWGSOP2 steps in elderly hemodialysis patients with functional, malnutrition-inflammation and comorbidity scales as these may be key causes and consequences of sarcopenia and the functional impact is most immediately felt by patients.

## 2. Materials and methods

### Ethics

This clinical investigation was approved by the ethics committee of the IIS- Fundación Jiménez Díaz UAM (act no. 03/19) and met the standards of the Declaration of Helsinki of the World Medical Association, as well as the Standards of Good Clinical Practice, in addition to complying with Spanish legislation on biomedical research (Act 14/2007). Participants signed an informed consent.

## Study design and subjects

This was a prospective study of outpatients on chronic hemodialysis from three Spanish external dialysis centers and a hospital dialysis unit of the Iñigo Álvarez de Toledo Renal Foundation performed in February 2019. The population was ethnically homogeneous (Caucasians). Inclusion criteria were clinical stability, age 75 to 95 years, able to perform physical condition assessment tests or dynamometry, a hemodialysis vintage of more than 3 months, which is the standard criterion to consider hemodialysis chronic, and signing the informed consent form. Exclusion criteria were failure to meet inclusion criteria; intra-dialysis instability; inability to perform the recommended tests (e.g. immobility or known neurologic condition that may have affected balance and gait regardless of muscle strength); conditions in which exercise is contraindicated, dementia and malignancy.

## Sarcopenia assessment

Primary variables were those in the 4-step EWGSOP2 (**Table 1**) and were assessed before the midweek dialysis session, except for BIA, which was performed during HD.

**A. Find.** Screening by the **SARC-F** survey score of Strength, Assistance walking, Rise from a chair, Climb stairs, and Falls [6].

**B. Assess.** Probable sarcopenia: loss of strength in any of the following tests.

1. Upper limbs: **Grip strength assessed by dynamometry (GSD)** using an electric dynamometer (CAMRY® Model EH101). The standing participant extends the arm parallel to the body without support or wrist movement and maintains maximal grip strength for 3 seconds, with 1-minute rest between each repetition, for two attempts by both arms. The best arm with the highest strength was used for the study.

2. Lower limbs: **sit to stand to sit 5 (STS-5)** evaluates the time required to get up from a chair five times without any support.

**C. Confirmation of sarcopenia.** **Appendicular skeletal muscle mass (ASM)** was assessed by a MALTRON® BioScan touch i8 bioimpedance device (BIA), during the second half of the mid-week hemodialysis session, as the device allows intra-hemodialysis measurements. The Maltron bioscan 916 device is validated for the measurement of body composition in situations characterizaed by changes in extracellular water (ECW) as this method monitors changes in ECW. Based on the equation: ECW = pL2/Re; during ultrafiltration ECW decreases while

**Table 1. Cut-off points for the diagnosis of sarcopenia in the 4-step EWGSOP2: Find-Assess-Confirm-Severity (FACS).**

| Step | Test | Male | Female |
|------|------|------|--------|
| *Find* | **SARC-F** (points) [6] | ≥4 | |
| *Assess* | **GSD** (kg) [10] | < 27 | <16 |
| | **STS5** (s) [11] | > 15 s for five rises | |
| *Confirm* | **ASM** (kg) [12] | < 20 | < 15 |
| *Severity* | ***GS*** *(m/s)* [12, 13] | ≤ 0.8 | |
| | **TUG** (s) [14] | ≥20 | |
| | **SPPB** (points) [15, 16] | ≤ 8 | |

SARC-F: Strength, Assistance walking, Rise from a chair, Climb stairs, and Falls, GSD: grip strength by dynamometry, STS-5: test sit to stand to sit 5, ASM: appendicular skeletal muscle mass, GS: gait speed, TUG: Timed-Up and Go test, SPPB: Short Physical Performance Battery.

extracellular resistance (Re) increases. This change in resistance is directly monitored by BIA. This software allows diagnosing dry weight when no further ECW is removed despite ongoing ultrafiltration and no changes in resistance are observed [17].

**D. Severity of sarcopenia.** Severity may be defined by any of the following variables:

1. **Gait speed (GS),** measured as time needed to walk 4 meters (also included as the 2nd test of the short physical performance battery, SPPB) expressed in meters per second, taking into account whether assistance was required to maintain balance during the walk (cane, walker, another hand). To make the test more accurate, acceleration and deceleration effects were removed by adding an extra meter at the start and at the end.

2. **The Timed-Up and Go test (TUG)** was used to evaluate agility and dynamic balance. The subject gets up from a chair, walks 3 meters, turns a safety cone, and sits down again. This test is performed at the maximum speed at which the patient can walk. The test is repeated three times and the shortest time is considered.

3. **The Short Physical Performance Battery (SPPB),** modified by Guralnik [18] consists of three tests. The first adapts the Romberg test for balance: patients are asked to stand, feet together, then instructed to place one foot next to the other, with the heel halfway down the other foot. Finally, feet should reach a tandem position, one foot heel in front of the toes of the other foot. In all three instances, preservation of balance is evaluated. This is followed by the GS and STS5 tests, as described above.

## Anthropometric, biochemical, and nutritional variables

Body mass index (BMI), Mid-Upper Arm Circumference (MUAC), waist-to-hip ratio (WHR), albumin (green bromocresol method), hemoglobin, normalized protein catabolic rate (nPCR) and 25 OH-vitamin D were assessed before the mid-week dialysis session. Dialysis efficiency was assessed by the Kt/Vurea and body composition by BIA.

## Functional scales

Malnutrition-inflammation score (MIS) [19], comorbidity (Charlson) [20], dependence (Barthel) [21] and fragility (FRAIL index) [22] were assessed.

MIS is a fully quantitative score adopted from the subjective global assessment. The Charlson comorbidity index is a simple, readily applicable method of classifying comorbidity through a weighted index that considers the number and severity of comorbid conditions and assesses the risk of death from comorbid disease. The Barthel Index is considered the best of the Activities of Daily Living measurement scales to assess dependence. The FRAIL scale consists of 5 components: (1) fatigue (feel tired all or most of the time); (2) resistance (difficult to walk 10 steps without rest and unaided); (3) ambulation (difficult to walk several hundred yards unaided); (4) illness (more than 4 of the following illnesses: hypertension, diabetes, cancer other than minor skin cancer, chronic lung disease, heart attack, congestive heart failure, angina, asthma, arthritis, stroke, and kidney disease.); and (5) loss of weight (more than 5%). The FRAIL scale gives a score ranging from 0 to 5, 1 point for each component, and categorizes people to be robust, prefrail, and frail if they score 0, 1 to 2, or 3 or above, respectively.

In addition, demographic (sex, age) and kidney disease (cause and dialysis vintage) variables were collected. Anthropometric measurements and physical tests were carried out by the same two Physical Activity and Sport Sciences specialists in all cases.

## Statistical analysis

Statistical analysis used the IBM SPSS Statistics V20 software. Quantitative variables were presented as mean and standard deviation and qualitative variables as absolute numbers and percentages. Student´s t test was used to compare quantitative variables between two groups. Correlations between variables were assessed by the Spearman rank test as multiple variables when non-normally distributed. Pearson was used to correlate numerical and categorical variables. Differences between qualitative variables were evaluated through the Chi-square test. The level of statistical significance was specified for $p < 0.05$.

## 3. Results

### 3.1 Baseline characteristics

Of 60 participants, 41 (68%) were men. Mean age was 81.85±5.58 years and dialysis vintage 49.88±40.29 months. Baseline data on demographic, anthropometric, analytical data and body composition are shown in **S1 Table**.

### 3.2 Prevalence of sarcopenia according to the new 4-step EWGSOP2

**Table 2**, presents the individual EWGSOP2 step results and **S2 Table** presents the same results split by gender. SARC-F found that 18/60 (30%) patients should be assessed for sarcopenia.

The sequential application of the four steps resulted in a prevalence of confirmed or severe sarcopenia of 5/60 (20%). The sensitivity of SARC-F for an eventual confirmed diagnosis of sarcopenia (sequential diagnoses of probable sarcopenia followed by confirmed sarcopenia) was 46% and the specificity 81%. The positive predictive and negative predictive values were 61% and 69%, respectively.

Given the low sensitivity of SARC-F for an eventual confirmed diagnosis of sarcopenia, we next explored steps 2 through 4 (from probable sarcopenia to severe sarcopenia), indepedently of the SARC-F result.

**Table 3** shows the estimates for sarcopenia prevalence according to each EWGSOP2 step. The prevalence of probable sarcopenia ranged from 75% when assessed by GSD to 88% when

**Table 2. Sarcopenia criteria and prevalence of sarcopenia according to the individual components of 4-step EWGSOP2 in very elderly hemodialysis patients.** Data presented as mean±SD or n (%), n = 60.

| | Individual criteria | Prevalence of sarcopenia |
|---|---|---|
| *Find* | | |
| **SARC-F** (points) | 2.6± 2.3 | 18 (30%) |
| *Assess* | | |
| **GDS** (Kg) | 19.2±6.6 | 45 (75%) |
| **STS5** (s) | 20.3±6.3 | 53 (88%) |
| *Confirm* | | |
| **ASM** (kg) | 19.3±3.8 | 24 (40%) |
| *Severity* | | |
| **GS** (m/s) | 0.69±0.27 | 42 (70%) |
| **TUG** (s) | 19.1±12.1 | 22 (37%) |
| **SPPB** (points) | 6.2±2.9 | 45 (75%) |

SARC-F: Strength, Assistance walking, Rise from a chair, Climb stairs, and Falls; GSD: grip strength by dynamometry, STS-5: sit to stand to sit 5, ASM: appendicular skeletal muscle mass, GS: gait speed, TUG: Timed-Up and Go test, SPPB: Short Physical Performance Battery. *$p < 0.05$ in bold.

**Table 3. Prevalence of sarcopenia according to the 4-step EWGSOP2 in 60 very elderly hemodialysis patients.** Data presented as n (%) representing prevalence in the n = 60 full population. **A)** Prevalence of severe sarcopenia according to 4-step EWGSOP2. **B)** Prevalence of fourth EWGSOP2 step results consistent with severe sarcopenia in the full study population.

| A) Severe sarcopenia according to 4-step EWGSOP2 | | B) Fourth EWGSOP2 step consistent with severe sarcopenia in the full study population | | |
|---|---|---|---|---|
| *Assess*: Probable sarcopenia | | | | |
| Criterion | Prevalence | | | |
| GSD | 45 (75%) | | | |
| STS-5 | 53 (88%) | | | |
| GSD and/or STS-5 | 58 (97%) | | | |
| *Confirm*: Sarcopenia confirmed by ASM | | | | |
| Criterion | Prevalence | | | |
| GSD+ASM | 23 (38%) | | | |
| STS-5+ASM | 22 (37%) | | | |
| GSD and/or STS-5+ASM | 24 (40%) | | | |
| *Severity*: Severe sarcopenia following confirmation of sarcopenia with the ASM criterion | | *Severity*: Severity results consistent with severe sarcopenia in the full study population | | *p value |
| Criterion | Prevalence | Criterion | Prevalence | |
| GSD+ASM+GS | 19 (32%) | GSD+GS | 34 (57%) | <**0.001** |
| STS-5+ASM+GS | 19 (32%) | STS-5+GS | 40 (67%) | <**0.001** |
| GSD and/or STS-5+ASM+GS | 19 (32%) | GSD and/or STS-5+GS | 40 (67%) | <**0.001** |
| GSD+ASM+TUG | 11 (18%) | GSD+TUG | 20 (33%) | <**0.001** |
| STS-5+ASM+TUG | 11 (18%) | STS-5+TUG | 22 (37%) | <**0.001** |
| GSD and/or STS-5+ASM+TUG | 11 (18%) | GSD and/or STS-5+TUG | 22 (37%) | <**0.001** |
| GSD+ASM+SPPB | 21 (35%) | GSD+SPPB | 36 (60%) | <**0.001** |
| STS-5+ASM+SPPB | 21 (35%) | STS-5+SPPB | 45 (75%) | **0.001** |
| GSD and/or STS-5+ASM+SPPB | 21 (35%) | GSD and/or STS-5+SPPB | 45 (75%) | **0.001** |
| GSD and/or STS-5+ASM+ GS and/or TUG and/or SPPB | 22 (37%) | GSD and/or STS-5+ GS and/or TUG and/or SPPB | 47 (78%) | **0.002** |

GSD: grip strength by dynamometry, STS-5: sit to stand to sit 5, ASM: appendicular skeletal muscle mass, GS: gait speed, TUG: Timed-Up and Go test, SPPB: Short Physical Performance Battery. *p<0.05 in bold.

assessed by STS-5 to 97% when considering either GSD or STS-5. By contrast, the prevalence of sarcopenia confirmed by ASM was lower: 40% of all patients met the ASM criterion and one of the probability criteria, indicating a prevalence of confirmed sarcopenia of 40% according to EWGSOP2 if the first "Find" step is dropped.

The prevalence of severe sarcopenia ranged from 18% to 35% of all patients when severity was assessed only in patients with sarcopenia confirmed by ASM, following the 4-step EWGSOP2 sequence. However, 33% to 78% of all patients had a "Severity of sarcopenia" step result consistent with severe sarcopenia, i.e. for every criterion or combination of criteria, the prevalence of functional impairment consistent with severe sarcopenia was 15% to 41% higher in the full population than indicated by the sequential 4-step EWGSOP2 (all p values ≤0.002).

## 3.3 EWGSOP2 and functionality, malnutrition-inflammation and comorbidity scales

Next, we assessed the correlation between EWGSOP2 and functionality, malnutrition-inflammation, and comorbidity scales. The Charlson comorbidity score was higher in men than in women (10.46±2.28vs 9.05±1.71, p = 0.020), but there were no gender differences in MIS (6.01 ±3.80 for all), dependence (Barthel 88.16+18.59 for all), or frailty (FRAIL 1.98+1.32 for all)

scales. Twenty-eight (47%) patients were at least moderately severe malnourished, 22 (47%) had at least moderate dependence, 19 (32%) were frail and 100% had high comorbidity according to the Charlson scale.

Table 4 describes the association of probable sarcopenia, confirmed sarcopenia and evidence of severe sarcopenia even when sarcopenia was not confirmed by BIA with Mis-nutrition, Barthel-dependence and Frail-fragility scales. Only evidence of severe sarcopenia, even if sarcopenia was not confirmed by BIA, correlated with the Barthel-dependence scale and approached significance with the Frail-fragility scale.

S3 Table presents the correlation between main variables and with other variables analyzed. There is a correlation between GSD and variables that indicate severity, but this correlation is not observed for ASM. The Find (SARC-F) and severity (GS, TUF, SPPB) steps correlated with the Charlson, MIS, Barthel and Frail scales, but not with anthropometric variables, while ASM correlated with the Barthel scale and with anthropometric variables (BMI, MUAC, WHR). GSD was the Assess test that correlated with more variables, both functional and anthropometric.

## 4. Discussion

Recently, EWGSOP2 established a consensus novel diagnosis and severity assessment for sarcopenia consisting of 4 sequential steps (4-step EWGSOP2), that we have now applied to persons over 75 years of age in hemodialysis. The main findings are the low sensitivity of the "Find" step for confirmed sarcopenia, the high prevalence of probable sarcopenia and the low prevalence of confirmed sarcopenia when the 4 steps in EWGSOP2 were sequentially applied, that contrasts with the high prevalence of evidence of dynapenia consistent with severe sarcopenia in the fourth step of the 4-step EWGSOP2. These findings raise questions as to the optimal assessment method for sarcopenia in elderly dialysis patients and raise the issue, to be addressed in future prospective multicenter studies, of whether a 2-step EWGSOP2 consisting of Assess-Severity may be simpler to apply and/or may provide better prognostic information than 4-step EWGSOP2 in elderly persons on hemodialysis. In this regard EWGSOP2 indicates that in clinical practice, probable sarcopenia is sufficient to trigger the evaluation of the causes and start interventions [5]. We recognize that this two-step procedure will assess dynapenia rather than sarcopenia. However, in the dialysis context, the parameter (sarcopenia or dynapenia) that better associates with outcomes should be identified and corrected.

The first issue raised is the need for the first "Find" step. As a screening step, it would require a high sensitivity, even at the expense of a compromised specificity. The low sensitivity of SARC-F for an eventual confirmed diagnosis of sarcopenia (46%) in elderly hemodialysis patients questions the need for this step in EWGSOP2. Indeed, a low sensitivity (25%) for the diagnosis of sarcopenia according to EWGSOP had previously been reported while the

**Table 4. Correlation between different sarcopenia categories and MIS Barthel and Frail indexes in 60 elderly hemodialysis patients.**

| Variable | | Probable sarcopenia | Confirmed sarcopenia | Severe sarcopenia according EWGSOP2 | Evidence of severe sarcopenia** |
|---|---|---|---|---|---|
| **MIS** | r | 0.145 | 0.069 | 0.063 | 0.150 |
| (points) | p | 0.268 | 0.600 | 0.630 | 0.252 |
| **Barthel** | r | -0.235 | -0.247 | -0.302 | -0.257 |
| (points) | p | *0.071* | *0.057* | **0.019*** | **0.048*** |
| **Frail** | r | 0.140 | 0.193 | 0.227 | 0.244 |
| (points) | p | 0.288 | 0.139 | *0.081* | *0.060* |

*p<0.05 in bold. Pearson correlation shown. Charlson was not assessed as all patients were in the highest category.

** independently from the results for muscle mass assessment.

specificity was 81.4% as referenced by the EWGSOP2 document [23]. If the first step is dropped, then the prevalence of confirmed sarcopenia raises to 40% in our population.

The second major finding is the higher prevalence of evidence of dynapenia consistent with severe sarcopenia than of confirmed sarcopenia. This is a striking finding that may have two non-mutually exclusive explanations: a) BIA is suboptimal to assess muscle mass in CKD; and b) In CKD dynapenia may be dissociated from sarcopenia. If the latter is the case, the next question would be which is a more clinically relevant parameter. Insight may be obtained from studies addressing their relationship of dynapenia or sarcopenia with outcomes. A dissociation was also observed between sarcopenia and dynapenia in younger hemodialysis patients (mean age 65 years) and dynapenia (without evidence of loss of muscle mass as assessed by creatinine index) was associated with an increased risk of death in adjusted analysis (HR = 2.99 p = 0.02) while sarcopenia was not [24]. The difference between muscle mass and strength found in our study may relate to poor muscle function despite acceptable muscle mass, as described in CKD patients [25, 26]. In CKD, uremic toxins may cause mitochondrial muscle dysfunction and calcitriol deficiency may facilitate muscle protein degradation [27].

Overall, the methods to confirm sarcopenia should be improved and adapted to routine clinical care needs. Magnetic resonance imaging (MRI) or computed tomography (CT) [5] are considered gold standard techniques but are not widely available and CT exposes to radiation, while Dual-energy X-ray absorptiometry (DXA) is marred by different DXA instruments brands not giving consistent results [5]. BIA is an accessible and inexpensive method for measuring muscle mass. Decreased muscle mass assessed by BIA is a good marker of mortality in patients with CKD [28, 29]. Additionally, BIA can be employed during the hemodialysis session, thus limiting the need for further healthcare visits in a population that already devotes at least 18–20 h per week to healthcare.

BIA does not measure muscle mass directly, but estimates muscle mass from whole-body electrical conductivity and may be influenced by the hydration status of patients [30]. Few studies have rigorously evaluated the best timing for body composition assessment in hemodialysis patients. Post-dialysis measures may result in lower lean tissue mass and intracellular water (ICW) estimates than pre-dialysis measures [31]. Assessing muscle mass at a consistent time relative to hemodialysis sessions has been suggested if repeated measures are planned, preferably 15 to 120 minutes post-dialysis, when patients are closer to their target weight [26]. However, the target weight is empirically determined, and it is uncertain whether post-dialysis weight represents the real weight. Finally, equations and algorithms originally developed in individuals without CKD are often used to estimate ASM [32]. The dynamics of hemodialysis units and the desire of patients to leave for home as soon as the thrice weekly session has ended makes it difficult to perform BIA post-dialysis.

It is worth emphasizing that the discrepancy between confirmed sarcopenia using BIA and evidence of severe sarcopenia is not limited to hemodialysis patients, as it has been observed in non-dialysis CKD patients. Thus, the technical limitations of BIA in hemodialysis patients do not appear to be the main issue or, at least, a dialysis-specific issue. In line with our results, a higher prevalence of severe sarcopenia than of confirmed sarcopenia was previously reported when using the 4-step EWGSOP2 in community dwelling older adults (6.0% vs 4.6%), when BIA was used to assess muscle mass [33]. The difference was accounted for by persons with CKD. In participants with eGFR above 60 ml/min/1.73 m2, the prevalence of severe sarcopenia was 4.7% and the prevalence of confirmed sarcopenia 4.9%, i.e. as expected, confirmed sarcopenia is more common than severe sarcopenia. However, in participants with eGFR below 60 ml/min/1.73 m2, the prevalence of severe sarcopenia was 10.3% and the prevalence of confirmed sarcopenia 3.6%. In another study in CKD stage 3–5 patients, sarcopenia prevalence was 10% when assessed by muscle function and 6% when assessed by muscle mass using BIA [34].

The present findings on correlations between sarcopenia and functionality, malnutrition-inflammation and comorbidity scales support that hemodialysis patients may present acceptable BIA-assessed muscle mass with poor functionality. Strength and variables that assessed severity correlated between them and with the scales of functionality, nutrition-inflammation and comorbidity, but not with muscle mass measured by BIA.

Some limitations should be acknowledged. The number of patients was not high as many of the tests are complex to perform by such an elderly population. However, this illustrates some of the limitations of EWGSOP2, as ability to perform physical condition assessment tests or dynamometry was a requirement for recruitment. Furthermore, DXA, MRI or CT were not used for sarcopenia confirmation. Again, our study represents routine clinical practice conditions and illustrates the difference between a research, high tech environment and the available resources in thousands of non-hospital-based dialysis units throughout the world. While clinical neuropathy was not diagnosed in any of the participants, and dialysis dose was adequate, based on $Kt/V_{urea}$ assessment, electrophysiologic studies were not performed to assess subclinical uremic polyneuropathy. Uremia is a systemic disorder, and we did not assess the potential contributors to the strength/muscle mass discrepancy. The study is focused on the older population (over 75) and the ≤75 years age group was not assessed. Thus, results cannot be extrapolated to younger patients.

## 5. Conclusions

According to the 4-step EWGSOP2, the prevalence of confirmed or severe sarcopenia was low (20%) in elderly hemodialysis patients. However, a dissociation was observed between the prevalence of severe sarcopenia according to the 4-step EWGSOP2 and the high prevalence of functional impairment consistent with severe sarcopenia (78%). Future studies should address whether a 2-step EWGSOP2 assessment (Assess-Severity) may be simpler and both increase feasibility and improve prognostic information in the routine clinical evaluation of elderly persons on hemodialysis, given the potential dissociation between poor muscle function and acceptable muscle mass in this population as well as the low sensitivity of the "Find" step.

## Supporting information

**S1 Table. Demographic, anthropometric, analytical data and body composition by bioimpedance.**
(DOCX)

**S2 Table. Sarcopenia criteria and prevalence of sarcopenia according to the individual components of the EWGSOP2 steps in very elderly hemodialysis patients divided by gender.**
(DOCX)

**S3 Table. Correlation between the parameters of the EWGSOP2 evaluation and other variables in 60 very elderly hemodialysis patients.**
(DOCX)

**S1 Data. Sarcopenia database.**
(XLS)

## Acknowledgments

We would like to thank the staff of the centers where the study was performed, especially Dr. Roberto Martín, Marcos García, Ismael Ballesta and Esther Jiménez for their invaluable assistance in the successful completion of this study.

**Institutional review board statement**

The study was conducted according to the guidelines of the Declaration of Helsinki, and approved by the Institutional Ethics Committee board of Fundacion Jimenez Díaz (act no. 03/ 19, February 2019).

**Informed consent statement**

Informed consent was obtained from all subjects involved in the study.

## Author Contributions

**Conceptualization:** M. Luz Sánchez-Tocino, Blanca Miranda-Serrano, Sebastian Mas-Fontao, Emilio González-Parra.

**Data curation:** M. Luz Sánchez-Tocino, Sebastian Mas-Fontao, Antonio López-González.

**Formal analysis:** M. Luz Sánchez-Tocino, Carolina Gracia-Iguacel, Sebastian Mas-Fontao, Antonio López-González, Alberto Ortíz, Emilio González-Parra.

**Funding acquisition:** M. Luz Sánchez-Tocino, Blanca Miranda-Serrano.

**Investigation:** M. Luz Sánchez-Tocino, Blanca Miranda-Serrano, Ana María de-Alba-Peñaranda, Antonio López-González, Silvia Villoria-González, Mónica Pereira-García.

**Methodology:** M. Luz Sánchez-Tocino, Blanca Miranda-Serrano, Ana María de-Alba-Peñaranda, Silvia Villoria-González, Mónica Pereira-García, Emilio González-Parra.

**Project administration:** M. Luz Sánchez-Tocino, Blanca Miranda-Serrano.

**Resources:** M. Luz Sánchez-Tocino.

**Software:** M. Luz Sánchez-Tocino, Sebastian Mas-Fontao.

**Supervision:** M. Luz Sánchez-Tocino, Blanca Miranda-Serrano, Emilio González-Parra.

**Validation:** M. Luz Sánchez-Tocino, Blanca Miranda-Serrano, Carolina Gracia-Iguacel, Sebastian Mas-Fontao, Alberto Ortíz, Emilio González-Parra.

**Visualization:** M. Luz Sánchez-Tocino, Blanca Miranda-Serrano, Carolina Gracia-Iguacel, Sebastian Mas-Fontao, Alberto Ortíz, Emilio González-Parra.

**Writing – original draft:** M. Luz Sánchez-Tocino, Blanca Miranda-Serrano, Carolina Gracia-Iguacel, Sebastian Mas-Fontao, Alberto Ortíz, Emilio González-Parra.

**Writing – review & editing:** M. Luz Sánchez-Tocino, Blanca Miranda-Serrano, Carolina Gracia-Iguacel, Sebastian Mas-Fontao, Alberto Ortíz, Emilio González-Parra.

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
