## [Decision Letter · Decision Letter 0]

14 Oct 2021

PONE-D-21-31006Sarcopenia assessed by 4-step EWGSOP2 in very elderly hemodialysis patients: feasibility and limitations.PLOS ONE

Dear Dr. Sanchez-Tocino,

Thank you for submitting your manuscript to PLOS ONE. After careful consideration, we feel that it has merit but does not fully meet PLOS ONE’s publication criteria as it currently stands. Therefore, we invite you to submit a revised version of the manuscript that addresses the points raised during the review process.

We look forward to receiving your revised manuscript.

Kind regards,

Pierre Delanaye

Academic Editor

PLOS ONE

Journal Requirements:

Additional Editor Comments:

Both reviewers question the fact the authors skip BIA measurement, as it is central in the diagnosis of sarcopenia. This point needs to be addressed.

The discussion is a bit too long and should be shortened, focusing on the results.

Reviewers' comments:

Reviewer's Responses to Questions

**Comments to the Author**

1. Is the manuscript technically sound, and do the data support the conclusions?

Reviewer #1: Yes

Reviewer #2: Partly

2. Has the statistical analysis been performed appropriately and rigorously? 

Reviewer #1: Yes

Reviewer #2: I Don't Know

3. Have the authors made all data underlying the findings in their manuscript fully available?

Reviewer #1: Yes

Reviewer #2: Yes

4. Is the manuscript presented in an intelligible fashion and written in standard English?

Reviewer #1: No

Reviewer #2: Yes

5. Review Comments to the Author

Reviewer #1: The reviewer has the following comments:

1. There are too many statistical analyses in this manuscript and, for most of them, they do not refer at all to the main objective of this study. For example, the authors compare men and women but this is completely useless and it confuses the main message of this paper. It is the same for the correlation between the parameters of the EWGSOP2. Once again, the authors must focus on the main objective of their research.

2. Focussing on their main objective, it could be interesting to assess the sensitivity, specificity, positive predictive and negative predictive values of the SARC-F questionnaire in their population. Indeed, the authors show that the SARC-F seems to miss out many real sarcopenic subjects and it could be interesting to compare the screening power of the SARC-F to what has been published in the scientific literature in a more general population.

3. The reviewer does not really understand how the authors could skip the bio-impedance assessment and still be able to assess a prevalence of severe sarcopenia since, by definition, the assessment of muscle mass is mandatory for the assessment of sarcopenia and its severity.

4. If the authors would like to skip the assessment of muscle mass, it is maybe because the problem of haemodialysed patients is not sarcopenia but more of muscle strength and muscle performance. It could deserve some discussion. Once again, the authors must focus on their main objective (i.e. to assess the prevalence of sarcopenia) and discuss all related items skipping all unrelated data clearly out of the field.

Reviewer #2: The paper is a study assessing the new EWGSOP sarcopenia criteria in hemodialysis patients. It is an interesting study, but important issues could be raised.

Major Comments

- I understand the idea of skipping the SARC-F questionnaire which is a screening tool in a high prevalence of sarcopenia population. Interestingly this screening tool “misses” a high number of patients with confirmed sarcopenia.

- However, I disagree that the muscle mass assessment could be skipped as discussed by authors. From the beginning of sarcopenia definition, muscle mass has been taken in account and what the authors propose in this large part of the paper is rather a study on dynapenia than sarcopenia. Sarcopenia is a state in which muscle strength is decreased, but also in which the muscle mass is decreased. Muscle plays a role in strength, but also has a metabolic role (for example of amino acid reservoir) and endocrine role (myokines secretion). Moreover, sarcopenia diagnosis is mostly driven by muscle mass as the authors explain (and as published in literature: see Bataille et al. The diagnosis of sarcopenia is mainly driven by muscle mass in hemodialysis patients. Clin Nutr 2016.). Therefore, the idea that sarcopenia frequency could be lower when muscle mass measurement is skipped is wrong. Although I understand that muscle strength is very closely related with outcomes, all papers on the consequences of sarcopenia on outcomes take in account the muscle mass. The end of paragraph on page 11 should be removed. I would also remove the Figure 1C and remove in Table 4 the part on Severity without ASM criterion.

- Instead, the authors could discuss the prevalence of sarcopenia using ESGWOP1 criterion.

-Table 5 could be removed as it does not provide many information since there is almost no difference between men and women. Results could be given in the text.

- Table 6: The table does not answer the main question which is: “Is EWGSOP2 sarcopenia definition correlated with MIS, dependance and frailty scores ?” The authors could provide this table as supplementary data and replace it by a table analyzing the correlation between sarcopenia using EWGSOP2 criterion (yes vs no), and malnutrition, dependance and frailty scores.

- The small number of patients, especially women included in this study is an important limitation.

- As already discussed by authors (page 20), the timing of BIA measurement is an important point. In the last KDIGO recommendations of 2020 on nutrition, the experts propose that the BIA measurement is done 30 min after the end of hemodialysis session. Could the author discuss this recommendation? Do the timing of measurement explain the limits of this technique to confirm sarcopenia in HD patients?

Minor comments:

Table 2: Fat mass (not Fast mass)

Table 4: EWGSOP (not EWSOP)

I would propose to replace “very elderly” by “elderly” in the title.

Could the authors provide de dosing technique used for albumin (Green bromocresol or Nephelemetry?). This could help interpreting the results.

6. PLOS authors have the option to publish the peer review history of their article (what does this mean?). If published, this will include your full peer review and any attached files.

Reviewer #1: No

Reviewer #2: No

---

## [Author Response · Author response to Decision Letter 0]

10 Nov 2021

Additional Editor Comments:

Both reviewers question the fact the authors skip BIA measurement, as it is central in the diagnosis of sarcopenia. This point needs to be addressed.

The discussion is a bit too long and should be shortened, focusing on the results.

As suggested by the reviewers and editor, we now separate the concepts of sarcopenia and dynapenia to address this criticism and have adapted the language (i.e. “fourth step assessment results consistent with severe sarcopenia” rather than “severe sarcopenia”).

At the editor's suggestion, the discussion has been substantially shortened.

Reviewer's Responses to Questions:

Reviewer #1: The reviewer has the following comments:

1. There are too many statistical analyses in this manuscript and, for most of them, they do not refer at all to the main objective of this study. For example, the authors compare men and women but this is completely useless and it confuses the main message of this paper. It is the same for the correlation between the parameters of the EWGSOP2. Once again, the authors must focus on the main objective of their research.

We appreciate the reviewer's comment and indeed the table presenting the population characteristics is too long. It has been simplified by eliminating variables that do not provide information in the description of the population. Furthermore, while results by sex are still present in tables, we skip this topic from the text and discussion. We leave results by sex in tables, as this will facilitate integration of data in future systematic reviews and this information is requested by major journals.

Correlation data have bene moved to suppl data. We respectfully believe that the fact that muscle mass did not correlate with the rest of the variables is important for the interpretation of results.

The results figure and former table 5 have been removed.

2. Focussing on their main objective, it could be interesting to assess the sensitivity, specificity, positive predictive and negative predictive values of the SARC-F questionnaire in their population. Indeed, the authors show that the SARC-F seems to miss out many real sarcopenic subjects and it could be interesting to compare the screening power of the SARC-F to what has been published in the scientific literature in a more general population.

We thank the reviewer for this this suggestion that indeed helps to visualize the results of the study. We now provide the sensitivity, specificity, positive predictive and negative predictive values of the SARC-F for confirmed sarcopenia as follows. “The sensitivity of SARC-F for an eventual confirmed diagnosis of sarcopenia (sequential diagnoses of probable sarcopenia followed by confirmed sarcopenia) was 46% and the specificity 81%. The positive predictive and negative predictive values were 61% and 69%, respectively.”

3. The reviewer does not really understand how the authors could skip the bio-impedance assessment and still be able to assess a prevalence of severe sarcopenia since, by definition, the assessment of muscle mass is mandatory for the assessment of sarcopenia and its severity.

As suggested by the reviewers and editor, we now separate the concepts of sarcopenia and dynapenia to address this criticism and use language that clarifies that indeed, assessment of muscle mass is mandatory for the assessment of sarcopenia and its severity. We additionally discuss recent publications that have also reported an (incongruous) higher prevalence of severe sarcopenia than of confirmed sarcopenia in the CKD population using 4-step EWGSOP2, i.e. they skipped the BIA for reporting prevalence without actually stating this clearly when reporting the results. 

4. If the authors would like to skip the assessment of muscle mass, it is maybe because the problem of haemodialysed patients is not sarcopenia but more of muscle strength and muscle performance. It could deserve some discussion. Once again, the authors must focus on their main objective (i.e. to assess the prevalence of sarcopenia) and discuss all related items skipping all unrelated data clearly out of the field.

As indicated above, we agree with the reviewer and now discuss separately sarcopenia and the more common evidence of dynapenia. 

Reviewer #2: The paper is a study assessing the new EWGSOP sarcopenia criteria in hemodialysis patients. It is an interesting study, but important issues could be raised.

Major Comments

- I understand the idea of skipping the SARC-F questionnaire which is a screening tool in a high prevalence of sarcopenia population. Interestingly this screening tool “misses” a high number of patients with confirmed sarcopenia.

- However, I disagree that the muscle mass assessment could be skipped as discussed by authors. From the beginning of sarcopenia definition, muscle mass has been taken in account and what the authors propose in this large part of the paper is rather a study on dynapenia than sarcopenia. Sarcopenia is a state in which muscle strength is decreased, but also in which the muscle mass is decreased. Muscle plays a role in strength, but also has a metabolic role (for example of amino acid reservoir) and endocrine role (myokines secretion). Moreover, sarcopenia diagnosis is mostly driven by muscle mass as the authors explain (and as published in literature: see Bataille et al. The diagnosis of sarcopenia is mainly driven by muscle mass in hemodialysis patients. Clin Nutr 2016.). Therefore, the idea that sarcopenia frequency could be lower when muscle mass measurement is skipped is wrong. Although I understand that muscle strength is very closely related with outcomes, all papers on the consequences of sarcopenia on outcomes take in account the muscle mass. The end of paragraph on page 11 should be removed. I would also remove the Figure 1C and remove in Table 4 the part on Severity without ASM criterion.

- Instead, the authors could discuss the prevalence of sarcopenia using ESGWOP1 criterion.

As suggested by the reviewers and editor, we now separate the concepts of sarcopenia and dynapenia to address this criticism.

We do agree with the reviewer that it is questionable to use a screening tool which has low sensitivity and have added discussion along these lines.

In table 4, Severity without ASM criterion has been removed and replaced by “Severity results consistent with severe sarcopenia in the full study population”

In agreement with the reviewer, the complete figure and its reference in the text of the results have been removed.

-Table 5 could be removed as it does not provide many information since there is almost no difference between men and women. Results could be given in the text.

Table 5 was removed and the description of the prevalence of comorbidity, undernutrition, dependence and frailty has is now in the text.

- Table 6: The table does not answer the main question which is: “Is EWGSOP2 sarcopenia definition correlated with MIS, dependance and frailty scores ?” The authors could provide this table as supplementary data and replace it by a table analyzing the correlation between sarcopenia using EWGSOP2 criterion (yes vs no), and malnutrition, dependance and frailty scores.

Table 6 has been moved to suppl, as suggested.

A new table 5 of correlations between sarcopenia EWGSOP2 YES/NO with nutrition, frailty and dependency is now presented. 

- The small number of patients, especially women included in this study is an important limitation.

The limitations of the sample size have now been acknowledged in the limitations section.

- As already discussed by authors (page 20), the timing of BIA measurement is an important point. In the last KDIGO recommendations of 2020 on nutrition, the experts propose that the BIA measurement is done 30 min after the end of hemodialysis session. Could the author discuss this recommendation? Do the timing of measurement explain the limits of this technique to confirm sarcopenia in HD patients?

Very appropriate remark. Indeed, the KDOQI nutrition guidelines recommend performing BIA in hemodialysis 30 minutes after the hemodialysis session as a tool for calculating body composition and especially muscle mass. However, it is not easy to find studies assessing muscle mass by BIA in hemodialysis patients that have performed the procedure after the hemodialysis session. This is understandable given the dynamics of hemodialysis units and the desire of patients to leave for home as soon as the thrice weekly session has ended. Two major recent studies performed BIA prior to the hemodialysis session and did not find an association of muscle mass with mortality (PMID 32830264, 28318630). 

Given the limitations of predialysis BIA and the feasibility issue for post-dialysis BIA, techniques have been developed to perform intradialysis BIA and we took advantage of one of these techniques. Intradialytic multifrequency BIA with Maltron bioscan 916 device is validated for the measurement of body composition in situations with changes in extracellular water (ECW). The principle of this method is to monitor changes in ECW. Based on the equation: ECW=pL2/Re; during ultrafiltration ECW decreases while extracellular resistance (Re) increases. This resistance change would directly be monitored by BIA. Thanks to this software, nephrologists can achieve dry weight when no further volume is removed from the ECW despite ongoing ultrafiltration and no changes in resistance are observed (15061473, 27928711). 

This information is reflected into methods and discussion.

Minor comments:

Table 2: Fat mass (not Fast mass)

Done

Table 4: EWGSOP (not EWSOP)

Done

I would propose to replace “very elderly” by “elderly” in the title.

Done

Could the authors provide de dosing technique used for albumin (Green bromocresol or Nephelemetry?). This could help interpreting the results.

Green bromocresol, added in the methodological section

---

## [Decision Letter · Decision Letter 1]

22 Nov 2021

PONE-D-21-31006R1Sarcopenia assessed by 4-step EWGSOP2 in elderly hemodialysis patients: feasibility and limitations.PLOS ONE

Dear Dr. Sanchez-Tocino,

Thank you for submitting your manuscript to PLOS ONE. After careful consideration, we feel that it has merit but does not fully meet PLOS ONE’s publication criteria as it currently stands. Therefore, we invite you to submit a revised version of the manuscript that addresses the points raised during the review process.

We look forward to receiving your revised manuscript.

Kind regards,

Pierre Delanaye

Academic Editor

PLOS ONE

Journal Requirements:

Additional Editor Comments:

The article has been improved and has the potential to be accepted. I agree with the discussion on sarcopenia/dynapenia. I agree with the Reviewer 1: all results according to gender have little added value (and are not discussed). At best, the results according to gender could be added in supplements. In the main text, I propose to focus on main results.

Reviewers' comments:

Reviewer's Responses to Questions

**Comments to the Author**

1. If the authors have adequately addressed your comments raised in a previous round of review and you feel that this manuscript is now acceptable for publication, you may indicate that here to bypass the “Comments to the Author” section, enter your conflict of interest statement in the “Confidential to Editor” section, and submit your "Accept" recommendation.

Reviewer #1: (No Response)

Reviewer #2: All comments have been addressed

2. Is the manuscript technically sound, and do the data support the conclusions?

Reviewer #1: Yes

Reviewer #2: Yes

3. Has the statistical analysis been performed appropriately and rigorously? 

Reviewer #1: Yes

Reviewer #2: Yes

4. Have the authors made all data underlying the findings in their manuscript fully available?

Reviewer #1: Yes

Reviewer #2: Yes

5. Is the manuscript presented in an intelligible fashion and written in standard English?

Reviewer #1: No

Reviewer #2: Yes

6. Review Comments to the Author

Reviewer #1: 1. The reviewer still believe that there are too many statistical analyses in this manuscript and, for most of them,

they do not refer at all to the main objective of this study (e.g. correlation between different sarcopenia categories and indexes).

2. The authors refer now to dynapenia. Again, this is not the objective of the paper.

Reviewer #2: The paper has significantly been improved with the provided modifications.

A typo error to be corrected: Pearson (and not Peasron) Page 8, line 187.

7. PLOS authors have the option to publish the peer review history of their article (what does this mean?). If published, this will include your full peer review and any attached files.

Reviewer #1: No

Reviewer #2: **Yes: **Stanislas BATAILLE

---

## [Author Response · Author response to Decision Letter 1]

29 Nov 2021

Additional Editor Comments:

The article has been improved and has the potential to be accepted. I agree with the discussion on sarcopenia/dynapenia. I agree with the Reviewer 1: all results according to gender have little added value (and are not discussed). At best, the results according to gender could be added in supplements. In the main text, I propose to focus on main results.

R: As suggested by the editor, we have now moved results by gender to suppl material. Additionally, the manuscript has been revised for English usage and typos.

Reviewer's Responses to Questions

Comments to the Author

1. If the authors have adequately addressed your comments raised in a previous round of review and you feel that this manuscript is now acceptable for publication, you may indicate that here to bypass the “Comments to the Author” section, enter your conflict of interest statement in the “Confidential to Editor” section, and submit your "Accept" recommendation.

Reviewer #1: (No Response)

Reviewer #2: All comments have been addressed

2. Is the manuscript technically sound, and do the data support the conclusions?

Reviewer #1: Yes

Reviewer #2: Yes

3. Has the statistical analysis been performed appropriately and rigorously?

Reviewer #1: Yes

Reviewer #2: Yes

4. Have the authors made all data underlying the findings in their manuscript fully available?

Reviewer #1: Yes

Reviewer #2: Yes

5. Is the manuscript presented in an intelligible fashion and written in standard English?

Reviewer #1: No

Reviewer #2: Yes

6. Review Comments to the Author

Reviewer #1: 1. The reviewer still believe that there are too many statistical analyses in this manuscript and, for most of them, they do not refer at all to the main objective of this study (e.g. correlation between different sarcopenia categories and indexes).

R: We are in a difficult situation regarding this comment, since the table that referee is 1 referring to was requested by referee 2, who now thinks that the manuscript is appropriate for publication as is. We cannot satisfy the request by referee 1 without failing to satisfy the request from referee 2. As suggested by the editor, we have now moved results by gender to suppl material.

2. The authors refer now to dynapenia. Again, this is not the objective of the paper.

R: We agree that this was not the objective of the study. However, the concept of dynapenia was introduced to address comments by the referees and we believe that it helps to understand and discuss the results. Our results, in line with those of the literature, suggest a greater impact of uremia on dynapenia than on sarcopenia (at least when muscle mass is assessed by tools available for routine clinical care). Since some criteria used by EWGSOP2 do in fact assess dynapenia, we believe that the distinction between reduced muscle mass (i.e. sarcopenia) and decreased muscle strength (i.e, dynapenia) is important for the discussion of the results, even if this was not the aim.

Reviewer #2: The paper has significantly been improved with the provided modifications.

A typo error to be corrected: Pearson (and not Peasron) Page 8, line 187.

Corrected

7. PLOS authors have the option to publish the peer review history of their article (what does this mean?). If published, this will include your full peer review and any attached files.

Do you want your identity to be public for this peer review? For information about this choice, including consent withdrawal, please see our Privacy Policy.

Reviewer #1: No

Reviewer #2: Yes: Stanislas BATAILLE

---

## [Editor Report · Decision Letter 2]

3 Dec 2021

Sarcopenia assessed by 4-step EWGSOP2 in elderly hemodialysis patients: feasibility and limitations.

PONE-D-21-31006R2

Dear Dr. Sanchez-Tocino,

We’re pleased to inform you that your manuscript has been judged scientifically suitable for publication and will be formally accepted for publication once it meets all outstanding technical requirements.

Kind regards,

Pierre Delanaye

Academic Editor

PLOS ONE

Additional Editor Comments (optional):

No further comments
---

## [Editor Report · Acceptance letter]

5 Jan 2022

PONE-D-21-31006R2 

Sarcopenia assessed by 4-step EWGSOP2 in elderly hemodialysis patients: feasibility and limitations 

Dear Dr. Sanchez-Tocino:

I'm pleased to inform you that your manuscript has been deemed suitable for publication in PLOS ONE. Congratulations! Your manuscript is now with our production department. 

Kind regards, 

on behalf of

Professor Pierre Delanaye 

Academic Editor

PLOS ONE